# Brain Metastasis Treatment: The Place of Tyrosine Kinase Inhibitors and How to Facilitate Their Diffusion across the Blood–Brain Barrier

**DOI:** 10.3390/pharmaceutics13091446

**Published:** 2021-09-10

**Authors:** Eurydice Angeli, Guilhem Bousquet

**Affiliations:** 1Institut National de la Santé Et de la Recherche Médicale (INSERM), U942, 9 Rue de Chablis, 93000 Bobigny, France; 2Assistance Publique Hôpitaux de Paris, Avicenne Hospital, Department of Medical Oncology, 93000 Bobigny, France; 3Sorbonne Paris Nord University, 99 Avenue Jean Baptiste Clément, 93430 Villetaneuse, France

**Keywords:** brain metastases, blood–brain barrier, tyrosine kinase inhibitors, pharmacokinetic

## Abstract

The incidence of brain metastases has been increasing constantly for the last 20 years, because of better control of metastases outside the brain, and the failure of most drugs to cross the blood–brain barrier at relevant pharmacological concentrations. Recent advances in the molecular biology of cancer have led to the identification of numerous molecular alterations, some of them targetable with the development of specific targeted therapies, including tyrosine kinase inhibitors. In this narrative review, we set out to describe the state-of-the-art in the use of tyrosine kinase inhibitors for the treatment of melanoma, lung cancer, and breast cancer brain metastases. We also report preclinical and clinical pharmacological data on brain exposure to tyrosine kinase inhibitors after oral administration and describe the most recent advances liable to facilitate their penetration of the blood–brain barrier at relevant concentrations and limit their physiological efflux.

## 1. Introduction

Brain metastases occur in 9 to 17% of all cancer patients [1], with a short survival [2,3]. Among the different cancer types, melanoma, breast and lung cancers are responsible for 6 to 56% of brain metastases [1].

Recent advances in the molecular biology of cancer have led to the identification of numerous molecular alterations, some of them targetable with the development of specific targeted therapies, including tyrosine kinase inhibitors (TKIs). For various cancer types, this new treatment era has considerably improved the prognosis for metastatic patients, with durable complete response [4,5,6]. This is particularly true for metastases that develop outside the central nervous system (CNS). In contrast, brain metastases, while usually sharing common gene alterations with extra-CNS metastases [7,8,9,10,11,12,13,14,15,16,17,18,19,20,21,22,23], are less sensitive to most anticancer agents. Indeed, diffusion into the brain parenchyma remains a pharmaceutical challenge because of the highly selective blood–brain barrier, combined with protective efflux systems limiting brain exposure to most xenobiotics [24].

Indeed, diffusion into the brain parenchyma remains a pharmaceutical challenge because of the highly selective blood–brain barrier, combined with protective efflux systems limiting brain exposure to most xenobiotics, two important topics that our research team has recently reviewed in [24,25].

Recently, next-generation TKIs have been approved for the treatment of metastatic breast cancer, lung cancer and melanoma, including patients with brain metastases, with promising response rates on CNS localizations. Thus, this topic has aroused growing interest, particularly over the last 5 years. 

In this narrative review, we set out to describe the state-of-the-art in the use of TKIs for the treatment of brain metastases, as well as recent advances liable to facilitate their penetration of the blood–brain barrier.

## 2. Tyrosine Kinase Inhibitors for the Treatment of Breast Cancer Brain Metastases

For the literature search on the Pubmed database, we applied the following algorithm: (“Tyrosine Kinase Inhibitor”) AND (“Brain Neoplasms” [MeSH] OR “Brain Metastases”) AND (“Lung Neoplasms” [MeSH] OR “Breast Neoplasms” [MeSH] OR “Melanoma” [MeSH]) (Figure 1).

Breast cancers are divided into three subgroups: (i) hormone-dependent breast cancers expressing estrogen and/or progesterone receptors; (ii) human epidermal receptor 2 (HER2) breast cancers with *HER2-neu* gene amplification and HER2 membrane receptor overexpression; and (iii) triple-negative breast cancers which overexpress neither hormone nor HER2 receptors. Among metastatic breast cancers, the incidence of brain metastases is much higher for the triple-negative or HER2 subtypes. For the HER2 subtype, since 2001, several anti-HER2-targeted therapies have been approved [26,27,28], providing a durable complete response for 30% of metastatic patients. However, other patients will systematically develop resistance to anti-HER2 treatment, and up to 37% of these patients will develop brain metastases and die because of the limited efficacy of drugs on these localizations [29,30]. There is little preclinical and clinical pharmacological data on the blood–brain barrier passage by anti-HER2 antibodies. When available, the cerebrospinal fluid-to-blood concentration ratio is usually very low, less than 1% [24]. In contrast, brain exposure is higher for anti-HER2 TKIs than for therapeutic antibodies, including lapatinib, neratinib and tucatinib. Given the difficulty in obtaining brain samples, it remains difficult to accurately assess drug concentrations in the brain parenchyma after intravenous administration, the most pharmacologically relevant method being the measurement of the free brain/free plasma ratio, called Kpuu [31]. Most studies still report the CSF/plasma concentration ratio, which probably overestimates the actual intracerebral concentration [32].

Lapatinib is an oral TKI reversibly blocking phosphorylation of the human epidermal growth factor receptor (EGFR) 1, HER2 and HER4, extracellular signal-regulated kinase 1 and 2 (ERK-1, 2), and protein kinase B (PKB/AKT). Lapatinib, combined with capecitabine, is currently approved for the treatment of metastatic breast cancer progressing after trastuzumab therapy [33] (Table 1).

In mice that developed brain metastases after intra-cardiac injection of HER2 cancer cell lines, concomitant intravenous injection of lapatinib significantly decreased the surface area of brain metastases, and this was associated with decreased HER2 phosphorylation of cancer cells [34]. In another murine orthotopic model of HER2 breast cancer brain metastases, after oral administration of (^14^C)lapatinib, the concentrations were seven to nine times higher in brain metastases than in normal brain parenchyma, suggesting greater diffusion across the altered blood–brain barrier. However, lapatinib concentrations remained five to ten times lower in brain metastases than in extra-CNS metastatic localizations [35]. Finally, various preclinical studies in mice have reported low CSF/plasma or brain/plasma ratios after lapatinib oral administration in mice [35,36,37,38] (Table 2). Even if the low molecular weight of lapatinib theoretically enables its brain penetration, lapatinib is quickly effluxed from the brain to blood via P-glycoprotein (P-Gp, coded by the *MDR1a/b* gene) and breast cancer resistance protein (BCRP, coded by the *BCRP1* gene) transporters. Using a triple knock-out mouse model for *MDR1a/b* and *BCRP1*, lapatinib brain concentrations were 40 times higher than in wild-type mice of the same genetic strain [39].

In patients, the benefit of using lapatinib alone for the treatment of HER2 breast cancer brain metastases is modest. In a pooled analysis of 12 studies and 799 patients, the use of lapatinib combined with capecitabine provided a brain response rate of 29.2% [40] (Table 1).

In a single pharmacological study, 12 patients with metastatic breast cancer had surgical resection of at least one brain metastasis after a single dose of lapatinib at 1250 mg given orally 2–5 days before surgery. Considerable variations in brain uptake of the drug were observed, between patients and within one and the same patient between different brain metastases [38] (Table 1).

Neratinib is an oral irreversible pan-HER TKI and a P-Gp inhibitor of low molecular weight. In a clinical phase III NEfERT-T study comparing the association of paclitaxel-neratinib with paclitaxel-trastuzumab for the first-line treatment of HER2 metastatic breast cancer, the neratinib arm was associated with a significant decrease in brain recurrence (8.3% versus 17.3%, *p* = 0.002) [41]. A phase II study was then conducted to specifically assess the benefit of using capecitabine and neratinib in the treatment of refractory HER2 breast cancer brain metastases. Thirty patients out of 49 achieved an objective response on brain metastases [42,43] (Table 1). To date, there is no preclinical pharmacological data for neratinib diffusion into the brain. In humans, one study reported brain and cerebro-spinal fluid pharmacological concentrations of neratinib in three patients after a daily dose of 250 mg for 7 to 21 days before surgical resection of HER2 brain metastases. At the time of craniotomy, cerebro-spinal fluid concentrations of neratinib were below the detection limit, while plasma concentrations were detectable. For one patient, the concentration of neratinib was measured in different parts of the resected brain metastasis, and neratinib concentrations were 1 to 10 times higher than plasma concentrations [44] (Table 2). 

Tucatinib is the latest generation of anti-HER2 TKI to be developed, and it has recently been approved for the treatment of HER2 metastatic breast cancer, including patients with refractory brain metastases.

In an orthotopic brain model of nude mice grafted with HER2 breast cancer, 75% of the tucatinib-treated mice were alive 22 days after grafting, whereas all animals were dead in the control groups treated with lapatinib alone or with placebo. In addition, in mice treated with tucatinib, there was a significant reduction in phosphorylated HER2 in the brain metastases, suggesting a direct inhibitory effect of tucatinib on HER2-overexpressing cancer cells [45].

In heavily pre-treated women with HER2 metastatic breast cancer, the HER2CLIMB trial demonstrated a considerable benefit of associating tucatinib with trastuzumab and capecitabine, with overall survival of 44.9% at 2 years [46]. In the sub-group of 291 patients with brain metastases, adding tucatinib reduced the risk of brain progression or death by 68% [47], with a median overall survival of 18.1 months (Table 1).

## 3. Tyrosine Kinase Inhibitors for the Treatment of Lung Cancer Brain Metastases

For this literature search on the Pubmed database, we applied the following algorithm: (tyrosine kinase inhibitor) AND (Brain Neoplasms [MeSH] OR brain metastases) AND (Lung Neoplasms [MeSH]) (Figure 1).

Lung cancer is the most common cancer in the world and the leading cause of death from cancer [48]. In the last two decades, targetable gene alterations have been identified in about 45% of patients with non-small-cell lung cancers (NSCLC) [49]. In particular, *EGFR* mutations or *ALK-EML4* translocations are identified in up to 50% of non-smokers or Asian patients with NSCLC of the adenocarcinoma histological sub-type [50].

NSCLC with *EGFR* mutations seems to be associated with an increased risk of brain metastases compared to patients with *EGFR* wild-type status [51]. In addition, the type of *EGFR* mutation could determine the phenotype of brain metastases, as exon 19 deletion is associated with more numerous small-sized metastases compared to exon 21 mutation and to *EGFR* wild-type genotype [52].

The first-generation EGFR TKIs, erlotinib and gefitinib, have been approved since 2005 for the treatment of metastatic NSCLC with activating *EGFR* mutations [53,54,55,56]. They are orally administered small molecules of less than 500g/mol kDa, reversibly inhibiting the adenosine triphosphate (ATP) binding site for EGFR tyrosine kinase, and thus the anti-apoptotic Ras signal transduction cascade. In vivo, preclinical pharmacokinetic studies of EGFR TKI distribution into the brain have shown a highly variable blood–brain barrier penetration of molecules, from 9 to 86% in normal mice [57,58] (Table 2). After oral administration, gefitinib brain concentrations are dose-dependent, reaching a peak at 1 h and rapidly decreasing as a result of P-Gp-mediated efflux [58]. In patients with NSCLC brain metastases treated with gefitinib, there is also dose-dependent cerebro-spinal fluid penetration, but it is very low, with a peak cerebro-spinal fluid /plasma ratio of 1.87% [59]. Significant brain metastasis responses have been obtained with erlotinib in lung cancers [60], up to 87% in a Japanese study [61] (Table 1).

The second-generation EGFR-TKIs, afatinib and dacomitinib, irreversibly bind to the tyrosine kinase domain of *EGFR*, and also to other ErbB-family members. Afatinib is an oral irreversible TKI that has been developed to specifically target *EGFR exon 19* deletion and *L658R* mutation. Afatinib treatment led to a significant progression-free survival improvement in patients with *EGFR*-mutated NSCLC, compared to platine-based chemotherapy (LUX-Lung 3 and LUX-Lung 6 clinical trials). A combined analysis of these two trials for the subgroup of patients with brain metastases showed improved progression-free survival (8.2 versus 5.4 months *p* = 0.03) [62]. In another study on patients with lung cancer brain metastases, afatinib provided a brain response rate of 81.1% [63] (Table 1).

Despite excellent CNS response rates for first- and second-generation TKIs, the prognosis of patients with *EGFR*-mutated lung cancer is linked to acquired resistance mutations, such as the *T790M* mutation [64]. After first- or second-generation TKIs, more than 30% of patients with NSCLC experience brain disease progression. Limited genomic data on surgically resected metastases has also evidenced acquired *EGFR* resistance mutations [64,65,66].

Osimertinib is a potent oral irreversible EGFR TKI developed to specifically target *EGFR T790M* resistance mutation. In preclinical murine models, osimertinib achieved a relevant brain exposure with a maximum brain-to-blood ratio of 2.2 at 60 min post-administration. Distribution was maintained in the brain up to 21 days after a single dose, osimertinib being then effluxed via P-Gp or BCRP blood–brain barrier transporters [67] (Table 2). In the phase III clinical trial FLAURA leading to FDA and European approval, osimertinib demonstrated its superiority compared to first- and second-generation TKIs [68,69]. A pooled analysis of data from 50 patients with brain metastases and *EGFR T790M* mutations reported a response rate of 54%, independent of prior brain radiotherapy [70] (Table 1). To our knowledge, there is no pharmacological data for osimertinib brain penetration in humans.

*Anaplastic lymphoma kinase* (*ALK*) gene rearrangements are associated with a high risk of brain metastases, occurring in up to 50–60% of patients in the course of their disease [71]. Crizotinib was the first ALK inhibitor approved for the treatment of metastatic *ALK*-mutated NSCLCs, with an improvement in survival compared to standard platine-based chemotherapy [72]. However, the intracranial efficacy of crizotinib is limited, because of its poor blood–brain barrier penetration. Three clinical cases report a cerebro-spinal fluid/plasma ratio of 0.06 to 0.3% after daily oral administration of crizotinib [73,74] (Table 2). After first-line treatment using crizotinib, all *ALK*-mutated NSCLC patients developed secondary resistance within 12 months, mainly due to acquired *ALK* mutations [75]. 

The second-generation ALK inhibitors alectinib and brigatinib were thus developed to efficaciously target these acquired *ALK* mutations. In addition, they have been associated with better survival results compared to crizotinib in first-line settings for metastatic ALK-mutated NSCLCs [76,77]. For brain metastases, alectinib provides high response rates ranging from 52 to 59% [78], and brigatinib provides even higher response rates of 78% [77] (Table 1).

More recently, lorlatinib, a third-generation *ALK* and c-ros oncogene 1 (*ROS1*), was developed for ALK-mutated NSCLCs harboring resistance mutations. During preclinical development, lorlatinib was optimized for a good cerebro-spinal fluid /plasma ratio [79] (Table 2). In a phase III study of 293 patients with ALK-positive metastatic NSCLC, lorlatinib was compared to crizotinib. At 12 months, 78% of the patients were alive in the lorlatinib groups versus 39% in the crizotinib group. Among patients with measurable brain metastases, 82% of the patients in the lorlatinib group exhibited intracranial response versus 23% in the crizotinib group. Seventy-one percent of the patients who received lorlatinib had complete intracranial response [80] (Table 1).

## 4. Tyrosine Kinase Inhibitors for the Treatment of Melanoma Brain Metastases

For the literature search on the Pubmed database, we applied the following algorithm: (tyrosine kinase inhibitor) AND (Brain Neoplasms [MeSH] OR brain metastases) AND (Melanoma [MeSH]) (Figure 1).

Among all cancers, melanoma is the third most common cause of brain metastasis [81]. About 50% of patients with cutaneous melanoma harbor the *BRAFV600E* hotspot mutation [82], which has led to the development of specific inhibitors, the first one being vemurafenib. In a phase II study of 90 patients with previously untreated brain metastases and *BRAFV600* mutation, treatment with vemurafenib yielded a brain response rate of 18% [83] (Table 1). More recently, combinations of BRAF inhibitors and MEK inhibitors (dabrafenib/trametinib, vemurafenib/cobimetinib, or encorafenib/binimetinib) have been approved for the treatment of patients with metastatic *BRAF*-mutated melanoma [84] but with little data on brain metastasis efficacy. In a prospective phase II study of 146 patients with brain metastases, monotherapy with a BRAF inhibitor (vemurafenib or dabrafenib) was compared to a combination of BRAF and MEK inhibitors (dabrafenib and trametinib). The median time to brain progression was not different between the two treatment arms, and it was under 6 months in both cases [85] (Table 1).

**Table 1 pharmaceutics-13-01446-t001:** Clinical trials on TKIs in different cancer types with brain metastases.

Cancer Type	Molecule	Sequence	CNS ORR (%)	PFS (Month)	OS (Months)	Trial Type	Reference
Breast cancer with HER2 amplification	Lapatinib	Lapatinib + capecitabine	29.2	4.1	11.2	MA	[40]
		lapatinib + capecitabine	20	6.4	NR	II	[86]
		Lapatinib	6	3.6	6.4	II	[86]
		Lapatinib + capecitabine	66	5.5	17	II	[87]
		Lapatinib + capecitabine	33	5.5	11	II	[88]
	Neratinib	Neratinib + paclitaxel	CNS PFS: NR	12.9	NA	III	[41]
		Neratinib + capecitabine	49	5.5	13.3	II	[42]
		Neratinib + capecitabine	33	3.1	15.1	II	[42]
		Neratinib + capecitabine	49	5.5	13.5	II	[43]
	Tucatinib	Tucatinib + trastuzumab + capecitabine	47.3	33.1	44.9	III	[46,47]
	Afatinib	Afatinib	70	NS	NS	II	[89]
		Afatitnib + vinorelbine	66	NS	NS	II	[89]
Lung cancer with EGFR mutation	Gefitinib	Gefitinib	87.8	14.5	21.9	II	[61]
	Erlotinib	Erlotinib + WBRT	86	-	-	II	[60]
		Gefitinib or erlotinib	-	vs. 10.2	vs. 31.6	II–III	[69,70]
	Afatinib	Afatinib	-	8.2	22.4	III	[62]
		Afatinib	81.1	-	-	R	[63]
	Osimertinib	Osimertinib	54	18.9	38.6	II–III	[69,70]
Lung cancer ALK-positive	Crizotinib	Crizotinib	12	-	-	I-II	[90]
		Crizotinib	23	39	NR	III	[80]
		Crizotinib	29	43	NA	III	[77]
	Lorlatinib	Lorlatinib	64	-	-	I–II	[90]
		Lorlatinib	82	78	NR	III	[80]
	Alectinib	Alectinib	54.2	9.6	-	III	[91]
		Alectinib	57	8.9	-	II	[92]
		Alectinib	75	-	-	II	[93]
	Brigatinib	Brigatinib	78	67	NA	III	[77]
BRAF mutated melanoma	Vemurafenib	Vemurafenib	18	-	8.9	II	[83]
		Vemurafenib	3.7	4	5.7	R	[85]
	Dabrafenib	Dabrafenib + trametinib	5.8 (median intracranial PFS)	7.3	11.2	R	[85]
		Dabrafenib	5.6	5.8	8.8	R	[85]

CNS Central Nervous System; ORR Objective Response Rate; PFS Progression Free Survival; OS Overall Survival; MA Meta-Analysis; IC Investigator’s choice; WBRT whole brain radiation therapy; R: Retrospective study; NR not reached; NA non available.

**Table 2 pharmaceutics-13-01446-t002:** Clinical and preclinical pharmacokinetic brain data for different TKIs.

Cancer	Drug	Molar Mass (g/mol)	Species	CSF/Plasma Ratio or Brain/Plasma Ratio	Time	Reference
Breast	Lapatinib	581	Mice	0.03	steady state	[39]
			Mice	0.02	2 h	[35]
			Mice	0.03	12 h	[35]
			Mice	0.05	AUC_0–16 h_	[37]
			Mice	0.04	AUC_0–16 h_	[37]
	Neratinib	557	Humans	<LOD	steady state	[44]
Lung	Erlotinib	393	Mice	0.01	1 h	[57]
			Rats	0.14	AUC_0–16 h_	[94]
	Gefitinib	447	Mice	0.01	1 h	[57]
			Mice	0.4	AUC_0–48 h_	[58]
			Mice	0.7	AUC_0–48 h_	[58]
			Humans	0.01	Day 30	[59]
			Rats	0.12	AUC_0–16 h_	[94]
	Afatinib	486	Rats	0.2	AUC_0–16 h_	[94]
	Osimertinib	163	Rats	6.09	AUC_0–16 h_	[94]
			Humans	3.8 (mean)	AUC_0-90 min_	[95]
	Crizotinib	450	Humans	0.001	5 h	[74]
			Humans	0.0006	5 h	[73]
	Lorlatinib	406	Rats	0.6	AUC_0-24h_	[79]
Melanoma	Vemurafenib	490	Humans	0.28–1.39	NA	[96]
			Mice	<0.1	4 h	[97]
	Dabrafenib	520	Mice	0.02	AUC_0-2h_	[98]
	Trametinib	615	Mice	0.1	AUC_0-48h_	[99]
	Cobimetinib	531	Mice	0.02	6 h	[100]
	Encorafenib	540	Mice	4.10–3	2 h	[101]

CSF cerebrospinal fluid; AUC Area Under the Curve; LOD Limit of Detection.

## 5. How to Facilitate Brain Diffusion of Anti-Cancer Tyrosine Kinase Inhibitors

To improve brain diffusion, drugs need to cross the blood–brain barrier without being effluxed from the brain to the blood. Several other approaches have been developed to increase drug penetration of the BBB [24] and therefore increase influx. In particular, the intranasal route of administration seems the most promising for small-sized molecules, and there is also promising data for the use of bi-specific antibodies of the angiopeps family.

### 5.1. Nanoparticle Formulation

Among different nanoparticles developed for blood–brain barrier penetration, most performant are polymeric nanoparticles, gold nanoparticles, and liposomes [102,103,104,105,106,107,108,109,110,111]. These nanoparticles have several advantages: (i) because of their small size, which normally does not exceed 100 nm, and their lipophilicity, they can cross the blood–brain barrier and accumulate at the tumor site as a result of the enhanced permeability and retention (EPR) effect; (ii) they can be loaded with different drugs. (iii) Finally, liposomes and polymeric nanoparticles ensure elimination of the product and limited toxicity because they are biodegradable [112,113].

In addition to these properties, several nanoparticles have been designed and engineered as “Trojan horses” to facilitate the crossing of the blood–brain barrier by TKIs. “Trojan horses” usually refer to bi-specific antibodies or systems using a ligand of physiological receptors located on the blood side of endothelial cells (reviewed in reference [24]).

In a study using murine models, lapatinib loaded on human serum albumin (HSA) nanoparticles had a 5-fold increased diffusion into the brain compared to lapatinib alone. This facilitated passage of the blood–brain barrier was explained by an increased passive diffusion of nanoparticles but also by active penetration using physiological albumin receptors [114]. Eight hours post-administration, the nanoparticle was almost absent from tissues. For gefitinib, two liposomal formulations have been tested using in vitro blood–brain barrier models. One formulation combined glutathione and polysorbate 80, known to be two ligands of physiological receptors in the blood–brain barrier, and another formulation used liposomes functionalized with RF, a cell-penetrating peptide. Both formulations showed an enhanced uptake across bEnd.3 cells, partially through clathrin-mediated endocytosis for RF-liposomes [115]. None of these molecules has been approved by the FDA to date.

### 5.2. Use of Ultrasound to Disrupt the Blood–Brain Barrier

This method combines the use of ultrasound with micro-bubbles. When excited by ultrasounds, micro-bubbles expand and exert a mechanical force on endothelial cells in the blood–brain barrier, thus disrupting tight junctions. This approach seems to not only have a mechanical effect but also to affect tight junction protein expression. Indeed, in murine models, mRNA and protein expression of CLAUDIN-5, OCCLUDIN, and ZONA OCLLUDENS-1 decreased in endothelial cells in the blood–brain barrier 3 h after low-frequency-focused ultrasound application. This biological effect of ultrasound could be explained by the tridimensional conformation change of tight junctions [116]. This approach has evidenced an enhanced brain penetration of drugs in preclinical and clinical studies [117,118,119]. For TKIs, one preclinical study explored the brain distribution of ^11^C-erlotinib in healthy rats after intravenous injection of micro-bubbles combined with brain application of ultrasound but was unsuccessful [120].

## 6. How to Reduce the Brain Efflux of Anti-Cancer Tyrosine Kinase Inhibitors

After diffusion into the brain parenchyma, physiological efflux systems expressed in the endothelial cells of the blood–brain barrier ensure the elimination of drugs from the brain to the blood. These protective efflux systems, by preventing drugs from reaching relevant and also potentially toxic concentrations in the brain parenchyma, also explain the limited efficacy of anti-cancer drugs on brain metastases. Among them, the ATP-binding cassette (ABC) transporters are ubiquitously expressed. They use energy from ATP hydrolysis to transport substrates across membranes. The most widely studied ABC efflux transporters at the blood–brain barrier are the P-glycoprotein (P-Gp), the breast cancer resistance protein (BCRP) and the multidrug resistance protein (MRP) [121].

All TKIs are substrates of ABC transporters, limiting their brain accumulation at relevant concentrations [99,122,123,124,125,126,127,128]. Using efflux transporter inhibitors might be a way to significantly enhance brain accumulations of anticancer drugs (reviewed in [25]), with promising preclinical data [97,129,130,131,132,133]. P-Gp inhibitors have failed in clinical trials. Indeed, next-generation inhibitors such as lorlatinib or osimertinib have been chemically engineered not to be recognized by the P-Gp-family efflux pumps.

## 7. Optimal Chemical Design of New Tyrosine Kinase Inhibitors

Lorlatinib was designed and structurally optimized in silico and in vitro for maximum brain penetration, and a minimum P-Gp efflux ratio [79]. Using in silico structure-based drug design, a macrocyclic template has been developed with a good brain penetration capacity. Then, using in vitro blood–brain barrier models with P-Gp-overexpressing cell lines, drugs have been optimized for a minimum efflux ratio. Typically, compounds that exhibited efflux ratios of under 2.5 were not retained, considered to have a low probability of achieving pharmacologically relevant brain concentrations in humans. A low molecular weight, high lipophilicity, and a low hydrogen bond donor count were independent factors for ratio performance.

The same methodological approach was used for osimertinib development [134], with a promising in silico prediction model of brain exposure [31,135], based on a quantitative structure–activity relationship (QSAR) model. Among 15 TKIs, osimertinib was identified to be the weakest substrate for blood–brain barrier efflux transporters in vitro, harboring fewer rotatable bonds for a more rigid molecule [136]. The free brain-to-plasma ratio (Kpuu) has been used to assess the BBB penetration by compounds and is the best predictor of brain penetration. It uses the in vivo brain-to-plasma ratio (Kp) combined with the in vitro determined fraction unbound in brain (fub) and in plasma (fup) (Kpuu = Kp × fub/fup). Among different compounds, all of those with an in vivo-estimated Kpuu > 0.3 were retained. In *cynomolgus* monkeys, ^11^C-osimertinib administered orally achieved a significant brain distribution from Positron Emission Tomography imaging [94].

## 8. Conclusions and Perspectives

Despite major improvements in the control of extra-CNS metastases, the treatment of brain metastases remains a pharmacological challenge due to the highly selective and protective blood–brain barrier. Different ways to overcome this limitation have been explored, such as nanoparticle formulations or the use of ultrasounds to disrupt the blood–brain barrier. Other possibilities have yet to be explored to enhance the delivery of TKIs to the brain at pharmacologically relevant concentrations and thus efficaciously treat brain metastases. In particular, one could imagine the use of the intranasal route of administration or the engineering of bispecific molecules combining TKIs with angiopeps. The main challenge remains the way in which to limit TKI brain efflux after brain penetration. Next-generation TKIs have been chemically designed to facilitate their penetration into the brain and to overcome efflux pumps, with promising clinical responses. These formulations need to be further improved.

## Figures and Tables

**Figure 1 pharmaceutics-13-01446-f001:**
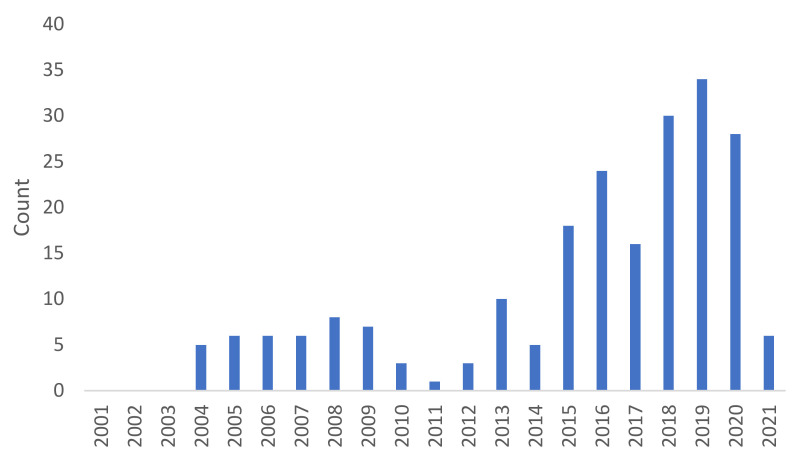
Publication trend since 2001 for kinase inhibitors and brain metastases from breast cancer, lung cancer and melanoma.

## Data Availability

Not applicable.

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
