# Peer review of "Brain Metastasis Treatment: The Place of Tyrosine Kinase Inhibitors and How to Facilitate Their Diffusion across the Blood–Brain Barrier"

_pharmaceutics, 2021, doi:10.3390/pharmaceutics13091446_

Round 1

Reviewer 1 Report

The review work titled “Brain Metastasis Treatment: The Place of Tyrosine Kinase Inhibitors and How to Facilitate Their Diffusion Across the Blood-Brain Barrier” by Angeli et. al. has been reviewed. The comments and suggestion are appended below.

  1. A figure showing publication trend in this field is required, e., number of publications per year and at least for the past 10 years or so to help readers understand the research engagement and the trend. This could be in the introduction section or in relation to the line numbers 40 to 42.
  2. Justify the need of this narrative review and that it is timely as there are various other closely related reviews available.
  3. Addition of a section on introduction of BBB, its construction, and the mechanism of how cancer and drugs crosses through it to the brain would be highly informative to the readers. Also, include how BBB limits entry of various drugs to treat the cancer in this context.
  4. The data presented in the section 2 are highly informative, presenting these data as a summary through a table or scheme would be more helpful to grasp the information easily.
  5. Line Nos. 118 – 120: Authors can provide a plot of the number of publications per year for this literature search as how much of work is available in this is not clear.
  6. Can a summary section or table for the section 3 be provided?
  7. “Table 1” needs to be referred in the text so that the text and table can be correlated.
  8. In Section, can the authors elaborate on what are the different types of nanoparticles used and what materials used for the surface modification of the nanoparticles gave the best results in terms of BBB permeation. Also, analyze whether the particles were toxic to the brain or biodegradable. What was the fate of these nanoparticles?
  9. Line No. 220: Authors have mentioned the size of nanoparticles that can diffuse through BBB is generally below 100 nm – add reference to this information.
  10. Section 5.2: Is there any other method tried for the BBB permeation?
  11. The review work does not present the future perspective, direction, or suggestions. This section should be added.
  12. The authors have presented various aspects of the brain metastasis treatment; however, the conclusion section has failed to summarize them. This should be improved.

Reviewer 2 Report

The manuscript entitled, “Brain metastasis treatment: the place of tyrosine kinase inhibitors and how to facilitate their diffusion across the blood-brain barrier,” (Pharmaceutics-1328050) by Angeli and Bousquet reviews the use of tyrosine kinase inhibitors to treat metastases in various adult malignancies. They include preclinical and clinical data on brain exposure to these drugs after oral administration and describe advances to increase CNS penetration by use of different formulations, disrupting the blood-brain barrier, or decreasing their brain efflux.

Comments

Page 9 of 14, line 57

What relevance does the CSF to blood concentration ratio have for a drug used to treat a tumor that grows primarily in the parenchyma of the brain? Unless data can be presented to show a correlation exists between the CSF to blood and tissue to blood ratios then it would seem to be meaningless.

Table 1

Inclusion of the molecular weight is not really necessary (especially not out to 3 significant digits).

Table 2

Some of the information contained within this table (CSF/plasma or brain/plasma ratio) can be found within the text of the manuscript (i.e., redundant) and should be removed; the concentration column is confusing and not very informative; and the “time” column is meaningless – thus, the table should be rethought or deleted.

Page 14 of 14 line 215

To improve brain diffusion, one could not only decrease efflux, but also increase influx – that wasn’t mentioned.

Page 14 line 240 and following

The use of ultrasound to improve CNS penetration has been discussed for many years (first study was in 2007 (Treat, Int J Cancer, 2007), and as the authors point out has had mixed success. It has not had sufficient success that it has become a clinically relevant method to increase the CNS penetration of poor penetrant TKIs.

Page 14 several paragraphs

Although the authors provide lapatinib and gefitinib as examples of drugs that can be formulated as nanoparticles or liposomes to improve their CNS penetration, these approaches have not worked sufficiently to be approved by the FDA-approved clinical products. So, at this point they really are just theoretical, and the authors should be realistic in their presentation of this point.

Pages 14 and 15

How to reduce brain efflux – the authors mention the use of efflux transporter inhibitors and that there is promising preclinical data, but in reality these compounds (e.g., elacridar, tariquidar, zosuquidar) have been evaluated extensively in clinical trials and have all failed. There’s a reason why there are no ongoing clinical trials with them. The authors should spend more time discussing the topic of the chemical engineering of compounds (e.g., osimertinib) to avoid efflux pumps and thus, increase the CNS penetration.

Round 2

Reviewer 1 Report

The authors have addressed the concerns and the work is now substantially improved. Hence, the work may be considered for publication.

Reviewer 2 Report

Although the authors have minimally addressed my comments in the revised manuscript.